# Adipose Fin as a Natural “Optical Window” for Implantation of Fluorescent Sensors into Salmonid Fish

**DOI:** 10.3390/ani12213042

**Published:** 2022-11-05

**Authors:** Yaroslav Rzhechitskiy, Anton Gurkov, Nadezhda Bolbat, Ekaterina Shchapova, Anna Nazarova, Maxim Timofeyev, Ekaterina Borvinskaya

**Affiliations:** 1Institute of Biology, Irkutsk State University, 664025 Irkutsk, Russia; 2Baikal Research Centre, 664003 Irkutsk, Russia

**Keywords:** implantable sensors, optical sensors, rainbow trout, skin transparency, tissue pH

## Abstract

**Simple Summary:**

Novel optical sensors require implantation into the most transparent organs in order to ensure the most reliable and rapid monitoring of animal health. Widely farmed salmonid fish, such as rainbow trout, have highly translucent adipose fin, which we tested here and showed its high potential as the implantation site for the fluorescent sensors. The filamentous sensors were convenient to inject into the fin, and their optical signal was easily detectable using a simple hand-held device even without immobilization of the fish. Responsiveness of the sensors inside the adipose fin to bodily changes was shown under induced acidosis of fish fluids. The obtained results characterize adipose fin as the favorable site for implantation of optical sensors into salmonids for real-time tracking animal physiological status in basic research and aquaculture.

**Abstract:**

Implantable optical sensors are emerging tools that have the potential to enable constant real-time monitoring of various internal physiological parameters. Such a possibility will open new horizons for health control not only in medicine, but also in animal husbandry, including aquaculture. In this study, we analyze different organs of commonly farmed rainbow trout (*Oncorhynchus mykiss*) as implantation sites for fluorescent sensors and propose the adipose fin, lacking an endoskeleton, as the optimal choice. The fin is highly translucent due to significantly thinner dermis, which makes the detectable fluorescence of an implanted sensor operating at the visible light range by more than an order of magnitude higher relative to the skin. Compared to the proximal parts of ray fins, the adipose fin provides easy implantation and visualization of the sensor. Finally, we tested fluorescent pH sensors inside the adipose fin and demonstrated the possibility of acquiring their signal with a simple hand-held device and without fish anesthesia. All these features will most likely make the adipose fin the main “window” into the internal physiological processes of salmonid fish with the help of implantable optical sensors.

## 1. Introduction

Various types of novel implantable sensors have the potential to revolutionize modern medicine and veterinary practices by providing the possibility of constant real-time monitoring of the physiological parameters that could only be previously measured with blood or tissue samplings [1,2]. Most of the described implantable sensors include electronic components and can utilize various methods for data transfer [1,3,4]. However, electronics-free alternatives are also available [5,6,7] and are much easier in terms of preparation for most biochemical laboratories. Such alternatives usually rely on light as the readout, and thus, acquisition of the sensor signal requires implantation into at least minimally translucent tissues.

Although most designs of implantable sensors are developed with humans and mammals in mind [5], other animals also deserve attention in this field. For example, fish aquaculture produces about half of the total amount of fish consumed by humans, and this market continues to grow [8,9]. Farmed fish suffer from infectious diseases [10] and decreases in water quality due to high crowding in the cages, during transportation [11], or due to sudden temperature increases caused by abnormal summer heat [12]. At the same time, the high stocking density of farmed fish makes it impossible to regularly control the status of each individual [13].

An implantable optical sensor typically consists of a sensing component (which varies in size from a single fluorophore molecule to large complexes involving proteins and microparticles) and a semipermeable polymeric carrier (typical size being from micrometers to millimeters, depending on the application) [5,14,15]. The carrier has two main purposes: (i) to minimize the immune foreign body reaction, which is important for delaying implant encapsulation, and thus keeping the contact with the interstitial fluid, and (ii) to specifically hide the sensing component within the carrier from adsorption of immune proteins (opsonins) that may potentially penetrate into the carrier and compromise the sensing part [16,17]. Therefore, although very different molecules of interest can be sensed [5], the upper size border of these molecules is generally defined by the lower size border of opsonins in the studied animal. The opsonins of teleost fish are similar to those of mammals [18], and most of them have a molecular weight of over 150 kDa. However, some fish lectins, which are considered to specifically recognize polysaccharides and glycosylated macromolecules, may be as small as 17 kDa [19]. These conditions generally restrict the detectable molecules of interest to low molecular weight substances; with exceptions that must be individually considered.

Physiological measurements using optical sensors usually rely on spectral changes of some type of luminescence (mostly fluorescence of the sensor), i.e., on tracking intensity ratios between two or more wavelengths or monitoring shifts of the spectral peaks [5]. For instance, the last principle was used to build an implantable sensor to the oncomarker human epididymis protein 4 with relatively low molecular weight [20]. Examples of optical sensors to complex non-protein molecules, such as the antibiotic kanamycin [6] or steroid hormones [21], now quickly emerge. However, most developed implantable optical sensors recognize more simple molecules, such as glucose [22], nitric oxide [23], or various ions [24,25,26]. Specifically, oxygen is now often measured via another approach with monitoring phosphorescence lifetime after pulsed excitation [14], which simplifies such technical challenges as photobleaching and tissue autofluorescence, in comparison to the spectrum-based approach. The oxygen sensors can also be successfully impregnated with various oxidases for indirect measurements of substances, such as glucose [27], lactate [28], or histamine [29], although the readout of those sensors can be affected by variations in the tissue oxygen level. Thus, the further development and cost reduction in the production and application of implantable optical sensors to key metabolites, hormones, and ions or signaling peptides may make these sensors one of the novel tools for automating animal health monitoring in veterinary practices and aquaculture.

Currently, examples of applications of optical sensors in fish are scarce. Previously, we managed to readily use an injected fluorescent sensor for tracking the interstitial pH of zebrafish (*Danio rerio*) [30], which are small in size and have thin skin. However, the latter is not the case for farmed fish species. Lee et al. were able to visualize a riboflavin sensor with infrared fluorescence under the skin of several fish species, but with limitations to some skin types of certain species [31]. However, tissues usually have increased transmission in infrared (over 730 nm) [32], and the sensors operating at the visible light range (400–730 nm) are expected to be less applicable in the same conditions. Thus, the selection of more translucent sites on the fish surface for each taxonomic group, or even individual species, can widen the number of optical sensors applicable in fish aquaculture.

Salmonids (Salmonidae, Chordata) are among the three main groups of globally farmed fish and include commonly cultured species, such as rainbow trout (*Oncorhynchus mykiss*) [9,33]. The most fortunate feature of salmonids for the application of implanted optical sensors is the presence of an adipose fin that stays seemingly translucent, even in adults. In this study, we chose rainbow trout to evaluate the potential of the adipose fin as the injection site for optical sensors in comparison to other parts of the fish. We used an implantable fluorescent pH sensor for these tests, since the concentration of protons is strictly regulated in the body fluids and changes can indicate the border of tolerance of certain animals to oxygen deficiency [34,35].

## 2. Materials and Methods

### 2.1. Preparation and Implantation of the pH Sensor

SNARF-1, a well-characterized pH-sensitive molecular probe, was used as the model sensing substance and was immobilized inside filamentous gel carriers. Protonated and deprotonated forms of SNARF-1 have different fluorescence spectra; hence, the intensity ratio between two wavelengths corresponding to these spectra indicates the ratio between the two forms and, mediately, the solution pH [36]. During the first stage, SNARF-1 conjugated with dextran (D-3304, Thermo Fisher Scientific, Waltham, MA, USA) was microencapsulated as described elsewhere [37]. Briefly, 2 mL of 2 mg/mL SNARF-1-dextran solution were sequentially mixed with 0.6 mL of 1 M CaCl_2_ and Na_2_CO_3_ solutions to precipitate the molecular probe into porous CaCO_3_ microcores. The cores were further covered with alternate layers of positively charged poly(allylamine hydrochloride) (Sigma-Aldrich, 283215, Burlington, MA, USA) and poly(sodium 4-styrenesulfonate) (Sigma-Aldrich, 243051, Hoeilaart, Belgium). Dissolving the CaCO_3_ cores in ethylenediaminetetraacetic acid solution gave us microcapsules, with the shell consisting of six layers of each polyelectrolyte and SNARF-1-dextran anchored inside.

The microcapsules were then embedded into a polyacrylamide hydrogel. The pH-sensing gels were prepared by mixing 30 μL of microcapsule suspension (~1.5 million microcapsules per 1 μL, measured in a hemocytometer) with 45 μL of water, 60 μL of the solution containing 30% acrylamide (A1089, AppliChem, Darmstadt, Germany) and 0.8% N,N′-methylenebisacrylamide (A3636, AppliChem, Germany), 1 μL N,N,N′,N′-tetramethylethylenediamine (Acros-13845, Acros Organics, Geel, Belgium), and 2 μL of 10% ammonium persulfate solution. Right after mixing, the solution was poured into smooth glass capillaries with an internal diameter of ~0.9 mm (Hirschmann, Baden-Württemberg, Germany) and was kept constantly rotating on the capillary axis in order to equally distribute the microcapsules inside the polymerizing gel. The obtained resilient (dense) gels carrying SNARF-1 were manually taken out of the glass capillaries and washed from unpolimerized components in excessive amounts of saline with antibiotic ampicillin (100 mg/L).

About 3-mm long resilient gels were implanted into fish tissues using 18G hypodermic needles (diameter 1.2 mm; SF Medical Products GmbH, Berlin, Germany). A steel wire was inserted into the needle and used as a plunger to push the sensor into the tissues after the puncture.

Specifically for testing the injections of amorphous (semiliquid) hydrogel into the dorsal fin, we reduced the proportion of acrylamide-bisacrylamide solution five times and polymerized it directly inside the syringe that was later used for injection with an insulin needle (0.3 mm outer diameter, BD Micro-Fine, Franklin Lakes, NJ, USA).

### 2.2. Acquiring Fluorescence of the Sensor and pH Measurements

We used three optical setups for different purposes: (i) visualizing the sensor fluorescence under tissue samples in equal imaging conditions; (ii) comparing its fluorescence spectra in tissue samples in vitro under equal illumination; (iii) acquiring its fluorescence spectra and respective pH measurements in tissues of rainbow trout in vivo.

The imaging setup included a dichroic mirror, a side-attached exciting 556 nm laser (FC-556–120 mW, CNI, Changchun, China) with a scatterer for equal illumination, a long-pass emission filter, and an EOS 1200D camera (Canon, Taichung City, Taiwan) with connected macro-lens (18–300 mm f/3.5–6.3 DC Macro OS HSM with EF-S, Sigma, Kawasaki, Japan). All presented fluorescence images were obtained with identically positioned optical components and samples, and with the laser set at maximum power in order to ensure the fluorescence intensities were comparable across the images. ISO was set to 100, and the exposure time was 30 s.

The presented fluorescence spectra were obtained using a Mikmed-2 microscope (LOMO, Saint Petersburg, Russia), with a QE Pro spectrometer (OceanOptics, Orlando, FL, USA; INTSMA-200 optical slit) attached to the photo-port, as previously described [37]. This setup included the same dichroic mirror and emission filter, but used excitation with a peak at 547 nm and included a 10× objective lens in the optical path. The microscope allowed us to quantitatively compare the intensity of sensor fluorescence in different tissue samples.

Finally, the third setup (the hand-held device; Figure 1) was specifically built for convenient pH measurements using the SNARF-1-based sensors inside the adipose fins of non-anesthetized rainbow trout. A dichroic mirror with a long-pass emission filter was mounted in the middle of a plastic T-pipe fitting. A 10× objective lens was attached to the lower end of the T-pipe, while F280SMA-A collimators (Thorlabs, Newton, NJ, USA) were mounted at two other sides of the pipe using kinematic adapters KAD11F (Thorlabs, Newton, NJ, USA). The 556 nm exciting laser was connected to the side collimator and the QE Pro fiber spectrometer (OceanOptics, Orlando, FL, USA) was connected to the upper collimator.

For the pH measurements using the third setup, the obtained spectra of SNARF-1 within the gels were later analyzed in Scilab v5.5.2 (ESI Group, Rungis, France), and their intensity ratios at 605 and 640 nm (I_605_/I_640_) were linearly calibrated in the range of phosphate buffers (pH from 5.5 to 8.8) prior to in vivo experiments [7]. Since the acrylamide gels required some time to equilibrate with the external pH, we started the measurements no earlier than 20 min after placing the sensor into any solution.

### 2.3. Obtaining Tissue Samples and Light Transmission Analysis

For the initial stages of this study, we used only recently deceased adult individuals of *O. mykiss* that were purchased from a shop located at a rainbow trout farm and transported to the laboratory in a thermostatic container at approximately 4 °C. Each purchase was made right after the scheduled fish delivery from the farm to the shop. Where necessary, tissue samples (skin, skin with the underlying muscles, the dorsal and adipose fins) were dissected with a scalpel and scissors, and kept in a saline solution during any procedures requiring more than several minutes. For some tests, the scales were scraped off from the skin samples with the scalpel. Brightfield images of the dorsal and adipose fins of rainbow trout were obtained using a Tough TG-5 camera (Olympus, Beijing, China) in microscope mode or using a macroscope Z16 Apo (Leica, Singapore) with an attached EOS 1200D camera (Canon, Taichung City, Taiwan).

We analyzed the light transmission through skin and the adipose fin in two ways: (i) by measuring spectra of transmittance (also known as the direct transmittance, as defined in [38]) using common spectrophotometry; and (ii) by comparing fluorescence intensity of the SNARF-1-based sensor under the tissues. Transmittance of tissues was analyzed using a CLARIOstar Plus microplate reader (BMG Labtech, Ortenberg, Germany) in 12-well plates. The results for each sample were averaged from ~100 measurements in the central part of the well (5 mm diameter). We also compared the optical density (synonymous to the negative decimal logarithm of transmittance represented as a fraction) of these samples, specifically at 650 nm. Muscle tissue was removed from skin samples as much as possible, and the adipose fins were longitudinally cut in half for these analyses. Rainbow trout show significant variations in coloring depending on the genetic line and growth conditions [39]. Nevertheless, skin in trout breeds with wild-type coloration can be classified into: dark skin on the dorsal part (virtually black in the case of fish used in this study), skin along the lateral line with pink shade (not pronounced in the used fish and further referred to as “pink”) and light skin on the ventral part (almost white in the used fish). Therefore, we included samples of each skin type for each individual to take into account their possible differences.

Additionally, in order to demonstrate the difference in the sensor fluorescence intensity due to the difference in transmittance between fish skin and the adipose fin, we applied the two optical setups as described earlier. The fluorescent microscope (the second setup) was used to acquire the spectra from the sensors implanted into the adipose fin and under the skin near the dorsal fin (dissected together with underlying muscle tissues) of the same individual. The microscope was also used to check the visibility of the fluorescent semiliquid hydrogel with SNARF-1 injected into the proximal part of the dorsal fin. The imaging setup allowed us to visually compare the fluorescent photos of the sensor inside the adipose fin and the sensor partially covered with fish skin (dark skin type).

### 2.4. Animal Handling and In Vivo Analyses

In order to demonstrate the in vivo functionality of the sensors inside the adipose fin, we monitored their readout for nine days and subjected the animals to experimental hypercapnia. All experimental procedures with live fish were conducted in accordance with the EU Directive 2010/63/EU for animal experiments and the Declaration of Helsinki; the protocol of the study was registered and approved before the start of the experiments by the Animal Subjects Research Committee of the Institute of Biology at Irkutsk State University (Protocol №2021/6). For the experiment, juvenile (0+) rainbow trout *O. mykiss* weighing 90–100 g were used. Before the start of the experiment, the fish obtained from the local farm were kept in a 300-L aquarium with constant aeration for 6 months in a cold room with a temperature of about 10 °C. The fish were fed with commercial feed Biomar No. 3 at the rate of 1.5% of body weight per day.

Immediately prior to the implantation procedure, individual sterile 18G hypodermic needles were prepared with inserted pH-sensitive gels. Fish (*n* = 10) were placed in an aqueous suspension of anesthetic clove oil (0.05 mL/L) until the animal lost its balance and ceased to respond to a pinch on the fin. Then, fish were transferred to the operation table and the adipose fin was flushed with a bactericidal 0.05% chlorhexidine digluconate solution. A needle loaded with the pH sensor was inserted about 0.6–0.7 mm into the tissue from the distal to the anterior edge of the adipose fin (at a depth of less than 1 mm under the skin). The gel piece was squeezed out with the steel wire and the needle was then removed from the flesh. Since the adipose fin of a juvenile is relatively small, this procedure placed the sensors in the middle between two dermal layers of the skin fold of the fin. The puncture hole was sealed with a small drop of Histoacryl glue (B. Braun, Barcelona, Spain) to prevent infection and protrusion of the gel from the tissue. Subcutaneous injections of the sensors near to the dorsal fin were also applied to a part of the animals in order to check the possibility of acquiring their signal from under the skin. To label individual fish, the dorsal or caudal fin of the fish was pierced with a sewing needle between the bone rays, and cotton threads of different colors treated with an antiseptic were tied. After that, the fish were immediately placed in a 300-L aquarium with aeration for recovery. Additionally, before the in vivo experiment, the same anesthesia procedure was applied to amputate the adipose fin of one juvenile fish that was used for checking the influence of the fin light transmission spectrum on the optical signal from the fluorescent sensors. For this, the sensor was equilibrated in the buffer with pH 6.6, and its readout was recorded using the hand-held device before and after covering the gel with the amputated adipose fin.

To record the signal from the sensor once a day (1, 2, 5–9 days after injection), the fish were transferred into 10-L plastic containers with cooled water (10 °C) and constant aeration. Then, the animal was carefully half-taken out of the water with a net, so that the fish was on its side and the head of the fish was under water. In this position, supported by the net, after several seconds of trying to escape, the trout usually demonstrated a freeze response for about a minute. Then, the operator supported the tail of the fish from below with one chilled hand, and with the other hand positioned the hand-held device near the surface of the adipose fin for signal registration (Figure 1b). To protect the eyes from the laser and facilitate the search for the gel, the operator put on glasses with a red-light filter; when the laser was aimed at the gel in the tissues of the fish, a red glow was visible through the glasses. Then, the laser was placed opposite the implant at about 5 mm from the skin, and the signal was recorded at several places in the gel (usually 12 shots). After that, the fish were immediately returned to their holding aquarium with aeration. The whole procedure usually took about 1–2 min.

To evaluate the responsiveness of the sensor to physiological pH changes, on the first (*n* = 10) and eighth (*n* = 7) days after sensor implantation, immediately after the signal registration at normal conditions, the fish were subjected to aqueous hypercapnia to induce metabolic acidosis [40]. For this, the fish were placed for 20–30 min in 10-L tanks with water saturated with CO_2_ (55–60 mg/L). Immediately thereafter, the signal from the implant was recorded in the adipose fin as described above. After this, the fish were returned to the aquarium with aeration and normal hydrochemistry for recovery. The level of CO_2_ in the water was determined using a CO_2_ test (NILPA, Moscow, Russia). All procedures with the fish were carried out at 10 °C. During the entire observation period, three fish lost the gel from the adipose fin, and two fish did not survive the second hypercapnic exposure; therefore, from the beginning to the end of the experiment, the signal was recorded in five individuals.

The statistical significance of the pH decrease after two hypercapnic exposures was verified using the Mann-Whitney U test with the Holm correction for multiple comparisons, as implemented in R v4.0.2 (R Core Team).

### 2.5. Animal Blood Collection and pH Measurements

Since widely used chemical anesthesia often depresses respiration and therefore enhances anaerobic processes and tissue acidosis [40], electrical stunning was chosen for instant immobilization of fish during the blood sampling procedure [41]. For this, on the fifth day after sensor implantation, immediately after the signal registration in the adipose fin, the fish were captured, hauled out of water, and placed on 12 × 6 cm steel electrodes (one under the head and one under the tail) coated with a conductive gel (133 g/L gelatin, 170 g/L NaCl) and immediately treated with electricity.

A Uni-T UTP1306 DC power supply (Uni-Trend Technology Limited, Hong Kong, China) was set to supply 25 mA (a limited voltage output of 32 V) in accordance with the recommendations for fish electronarcosis [41]. Blood was rapidly collected from the caudal vein of the immobilized fish by a syringe with a heparinized 16G hypodermic needle (SF Medical Products GmbH, Berlin, Germany). The current was then turned off and the fish were immediately returned to their tanks for recovery. The pH of approximately 200 µL of freshly collected blood was immediately evaluated using a Seven2Go Pro pH-meter with a glass microelectrode (Mettler Toledo, Columbus, OH, USA). The whole procedure took no more than 1 min.

The blood plasma was additionally separated by centrifugation at 5300× *g* for 30 s in order to calibrate the sensor readout for the possible influence of plasma components onto the calibration curve [42]. For this procedure, acquisition of the sensor fluorescence using the hand-held device was made in parallel with the plasma pH measurement using the pH-meter with the microelectrode.

## 3. Results and Discussion

### 3.1. Testing Sites for the Implantation of Optical Sensors

The potential sites for the implantation of optical sensors on the surface of an animal must be considered from the following perspectives: (i) the possibility to easily implant the sensor and keep it at the site for a long time; (ii) the possibility to acquire the optical signal from the sensor with the available technical solutions; and (iii) the responsiveness of the tissues in the site to changes of the parameter of interest in the whole organism. We tested injections of filamentous hydrogels (diameter ~0.9 mm; about 3 mm long) into different parts of recently deceased adult individuals of rainbow trout (Figure 2a). The gels retain their form and can carry a significant amount of the fluorescent sensing component, which should simplify their detection; importantly, they were found to be convenient for the implantation. Our overall experience shows that a substantial amount of flesh at the chosen site and a deep injection are required; otherwise the gels will be immediately pushed out by the tissues. Additionally, equilibration of the sensor internal media with interstitial fluid requires time, and a more significant amount of tissues around the sensor will induce a faster optical response after physiological changes. This condition also favors more fleshy sites.

The most obvious choice is subcutaneous implantation of optical sensors under the skin covering the muscles of rainbow trout, especially along the muscular region of the dorsal part and near the caudal fin. We tested the injections near the caudal and dorsal fins, and the procedure was quick and simple. The skin on the fish head seems to be more translucent than on the main body but it is tightly fixed to the head bones; hence, subcutaneous injections of sensors are not straightforward in this case. Transparent adipose eyelids are used for the implantation of various tags in salmonid marking [43], but any illumination near fish eyes for fluorescence excitation would cause additional anxiety, and is therefore not recommended during physiological measurements. Furthermore, animal anxiety would cause additional complications during signal acquisition.

Rainbow trout have a number of fins, which are more visually translucent than the skin on the main body. The proximal parts of the ray fins, especially the dorsal fin (Figure 2b), have a substantial amount of connective tissue; however, in this case, the sensor can only be observed from one side of the fish. These sites are also used for fish marking with tag injections [44]. We were successful in the implantation of the resilient gel into the proximal part of the dorsal fin of adult fish. However, the procedure may be less straightforward in the case of juveniles with a smaller dorsal fin, since the rays can interfere with the injection. We also tested injecting an amorphous hydrogel that had previously been found useful as the sensor polymeric carrier inside small fish [30]. Such an injection into the proximal part of the dorsal fin requires a smaller needle; it was indeed found to be easier and still successful. However, since the sensors are generally not intended for human consumption, fins will probably have to be amputated before selling the fish, which may jeopardize the marketable condition of the farmed fish if it is one of the ray fins.

The adipose fin (Figure 2c), which is structurally similar in different salmonids and does not contain an endoskeleton [45,46], lacks all of these drawbacks of other fins. It was found fleshy enough to easily implant the gel, compact enough to quickly find the gel from either side, and it has rich and loose connective tissue. The fin is extensively innervated and seems to serve as a mechanosensing organ during swimming [47], which probably implies good blood supply. Importantly, it can be removed without affecting the marketable condition of the fish, as is already practiced in some countries for labeling hatchery salmonids [47].

Despite its name, the salmonid adipose fin does not contain adipose tissue and is mainly supported by rods of collagen [45]. Transversal sectioning of the adipose fin (Figure 2d,e) demonstrated the important differences between skin on the fish back and on the fin. The upper skin layer (epidermis) is similar in both cases, but the underlying layer (dermis) has a structural shift at the basement of the adipose fin. In the skin on the body, the dermis consists of dense bundles of regular collagen fibers, but in the skin of the adipose fin its structure becomes rarer and more translucent. The free edge of the adipose fin is formed by radially-arranged spear-shaped collagen polymers, known as actinotrichia, which provide the fins with physical support. The actinotrichia are oriented perpendicular to the surface of the adipose fin, which is probably the reason why light penetrates deep into the fin. Furthermore, the rest of the connective tissues inside the fin (hypodermis and collagen matrix) are also markedly translucent. Transections of the dorsal fin (data not shown) showed a similar shift in the dermis structure, which confirms the expected potential of the proximal parts of the ray fins for the implantation of optical sensors in certain cases.

Therefore, although the adipose fin seems to be the natural choice for the implantation of optical sensors into salmonid fish, the proximal parts of the other fins can still be considered for the taxonomical groups lacking an adipose fin.

### 3.2. Light Transmission through the Skin and Adipose Fin

The proximal parts of the ray fins of rainbow trout are hard to cut for transmission analysis without damaging, so we further concentrated on the comparison of the optical properties between the skin and the adipose fin. For this purpose, we obtained transmittance spectra of each skin type (dark from dorsal, “pink” from lateral and light from ventral parts of the fish) and longitudinally cut halves of the adipose fin of recently deceased animals. Adult individuals were chosen for the analysis in order to obtain data for the thickest tissues with the lowest transmittance possible. The overall results are compared in Figure 3a.

Excitation of sensor fluorescence is preferred with the longest wavelengths possible, since ultraviolet (below 400 nm) is toxic [48] and blue light (roughly 400–500 nm) has the potential to cause an imbalance in mitochondria and cell damage [49]. Some deleterious effects have been reported at least up to turquoise blue of about 480 nm [50]. Fish skin and adipose fin of adult rainbow trout have similar transmission in the near ultraviolet (the measurements started from 350 nm) and are comparable in this regard up to violet blue (Figure 3a).

However, the adipose fin starts to excel skin at least by the factor of two from 420 nm, and keeps this advantage until the end of the measured wavelengths range at 1000 nm. Moreover, the difference is even more dramatic in the range with no known phototoxicity: between 515 and 700 nm (and at some wavelengths in infrared over 730 nm) the average transmission of the adipose fin is at least four-fold more stable and higher. At the range 530–545, it outgoes skin by five times.

We further individually compared the light transmission of the adipose fin and each skin type at 650 nm. We also analyzed skin types with removed scales (Figure 3b). The comparison demonstrated no substantial difference in the optical density between black skin on the back, slightly pink skin around the lateral line, and white skin on the abdomen. The removal of scales did also not result in a dramatic increase in skin transmission.

It is important to highlight that the tissue transmittance spectra and their individual variability can directly influence the readout of ratiometric optical sensors. Non-horizontal (i.e., showing dependence on light wavelength) transmittance spectra, such as those obtained for rainbow trout tissues (Figure 3a), may give the following effects: (i) shift the sensor calibration curve due to permanent influence on the fluorescence intensity ratio; and (ii) increase the measurement error because of individual spectrum variability. Thus, these effects must be considered for spectrum-based measurements with implanted optical sensors.

### 3.3. Visibility of the Fluorescent Sensor Inside Fish Tissues

To demonstrate the difference in the effectiveness of fluorescence acquisition between skin from the back of fish and the adipose fin, we chose pH-sensitive SNARF-1 as the model molecular probe. Both excitation (~560 nm) and emission (~590–640 nm) peaks of this dye lay in the range of four-fold higher transmission of the fin in comparison to the skin (Figure 3a). pH is also an important marker of the tissue status under oxygen deficiency [51], and may be monitored during fish transportation [11]. Resilient gels carrying SNARF-1 were implanted in amputated parts of recently deceased fish or covered with skin of the same individual for comparability.

Under the fluorescent microscope, we acquired several spectra from the sensors implanted into the adipose fin and between skin and muscles. Comparison of two spectra with maximal intensities in each injection site (Figure 4) showed about 18 times higher fluorescence intensity in the case of the adipose fin, which is close to the 20-fold value predicted from the difference in transmission at the excitation and emission wavelengths (Figure 3a). It should also be noted that the fluorescence of the sensor under the skin was barely visible through eyepieces, while the sensor inside the adipose fin was easy to find under the microscope. Thus, the sensor signal is potentially measurable from both implantation sites, but the adipose fin provides the opportunity to substantially reduce the amount of expensive molecular probe within the gel carrier. Additionally, we checked the visibility of SNARF-1 within the semiliquid hydrogel injected into the proximal part of the dorsal fin. Since the carrier was amorphous, the results were not quantitatively comparable with the data for resilient gels. However, it showed good visibility of the sensor fluorescence intensity, which was seemingly close to one of the gels inside the adipose fin. Thus, the bases of the ray fins are indeed a viable option for the application of optical sensors, even if injections of resilient gels would be challenging in certain cases.

We used the fluorescence intensity ratio at 605 and 640 nm I_605_/I_640_ to measure pH using the SNARF-1-based sensors. The average transmittance spectrum of the adipose fins of adult rainbow trout (Figure 3) showed approximately 1.25 better transmission for 640 nm than for 605 nm. This was a significant difference that could have substantially shifted the sensor readout and must be considered. However, individual variability was found to influence this value by no more than 7%, which lies within the measurement error range of the sensor itself. It gives the opportunity to simply modify the sensor calibration curve (usually prepared using buffer solutions in vitro) and obtain more correct absolute pH values for in vivo measurements if necessary.

### 3.4. Testing the pH Sensor Inside the Adipose Fin

Finally, since the adipose fin of rainbow trout is a peripheral organ, its responsiveness to physiological changes on the organism level should be tested. Acquisition of the optical signal from an implanted sensor with a stationary microscope is hardly feasible using live fish without anesthesia. In order to test in vivo functionality of the pH sensor inside the adipose fin, we built a hand-held optical setup that can be directly used in the chamber with fish, and partially immersed into water if necessary. The device included a 10× objective lens with two attached optical fibers for fluorescence excitation and spectral acquisition using an external laser and a spectrometer, respectively (Figure 1). This setup gave us the freedom of movement required for experiments on active fish.

For the in vivo experiment, we chose juveniles of rainbow trout as their tissues are thinner and should be more translucent, which in turn means less significant influence on the signal of the ratiometric pH sensors. Indeed, comparing the readout of the free sensor in a buffer with and without covering the gel with the amputated juvenile adipose fin demonstrated no influence of the 2-mm thick fin. Additionally, verification of the sensor readout inside extracted fish blood plasma also showed no need for adjusting the calibration curve, as has been required in some cases [42].

Unfortunately, the sensitivity of the hand-held device was found to be lower in comparison to the microscope and was not enough to acquire the spectra of the SNARF-1-based fluorescent sensors under the skin of rainbow trout. Thus, we were unable to compare the time of response of the sensor inside the adipose fin and under the skin to changes in blood pH. However, the spectral acquisition from the adipose fin was relatively straightforward, and usually required no more than 2 min per individual. It should also be noted that the injections of the gels into the adipose fin of juvenile rainbow trout were not more complicated in comparison to adult fish.

Overall, the monitoring of tissue pH inside the adipose fin for nine days (Figure 5) demonstrated a relatively high median pH of ~7.9 one day post injection, followed by a slow decrease and stabilization at the median level of 7.3–7.4 five days after the injection. Since the median blood pH of 7.2 (Figure 5) for these individuals was measured only on the fifth day of the experiment, the tissue alkalosis during the first days after the injection can be explained both by systemic and local pH increases. In the former case, it could be the consequence of the applied anesthesia, which may provoke systemic hypoxia-mediated respiratory alkalosis [52]; in the latter case, it may reflect a local hypocapnia in the wound area [53]. Thus, this effect requires further investigation.

Importantly, the tissue pH clearly reacted to the systemic acidosis caused by the elevated water CO_2_ level (Figure 5), both on days 1 and 8 post injection. Tissue pH decreased by 0.55 and 0.4 from the respective starting points (both *p*-values < 0.01) and recovered over the following days. Comparable changes in blood pH were observed in previous studies under hypercapnic and hypoxic conditions, as well as due to significant temperature changes [54,55]. Thus, the performed experiment demonstrates the functionality of the prepared hand-held device for repetitive pH measurements on the same rainbow trout individuals, and sensitivity of the sensor inside the adipose fin to physiological changes during at least the first days after implantation. More prolonged monitoring may face challenges posed by the immune foreign body response and sensor photobleaching, and these potential limitations should be studied further.

It must be noted that the sensor readout was verified only in fish plasma, but not in interstitial fluid. We cannot fully exclude the possibility that some difference in these two media may affect the absolute pH values measured using the proposed technique. Furthermore, in adult individuals the sensor readout would be more influenced by the tissue transmittance spectrum, and this would also shift the absolute pH measurements. Nevertheless, these factors should not affect dynamic monitoring of pH changes under various conditions.

The proposed approach involving the adipose fin as a unique site for the implantation of fluorescent sensors can already be used in physiological research on salmonids for pH monitoring without anesthesia. However, the adipose fin has significant potential for the future application of other optical sensors and could be further transferred to fish aquaculture. The conditions of fish farms would naturally require a very different device for signal acquisition from each individual fish. In the simplest cases, the device could be implemented as a part of automatic fish counters, currently used for high-throughput biomass monitoring on farms.

## 4. Conclusions

The proximal parts of the ray fins demonstrate a promising alternative to the implantation of optical sensors under the skin of the main body for large fish species. However, more convenient options can be found for certain taxonomical groups, as we demonstrated here for salmonids. The adipose fin of rainbow trout was shown to be transparent enough to easily detect fluorescence from the sensor operating at the visible light range and was much better suited for this purpose compared to fish skin. Moreover, it is small enough to quickly find the sensor from either side of the fish, and large enough for convenient implantation. We believe that all these features will make the adipose fin the “optical window” into internal physiological processes of salmonids with the help of implantable fluorescent sensors, both for solving problems of fundamental research and in aquaculture.

## Figures and Tables

**Figure 1 animals-12-03042-f001:**
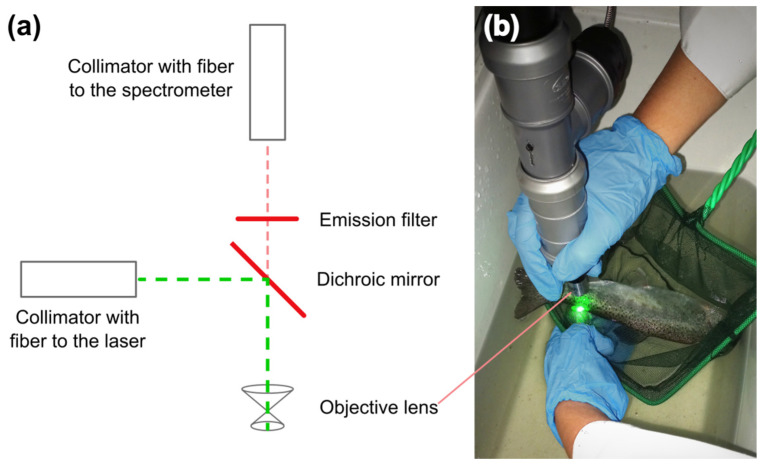
The optical setup (hand-held device) for in vivo pH measurements using implanted sensors. (**a**) Optical scheme of the setup. (**b**) Photo of the device during pH measurements on non-anesthetized fish.

**Figure 2 animals-12-03042-f002:**
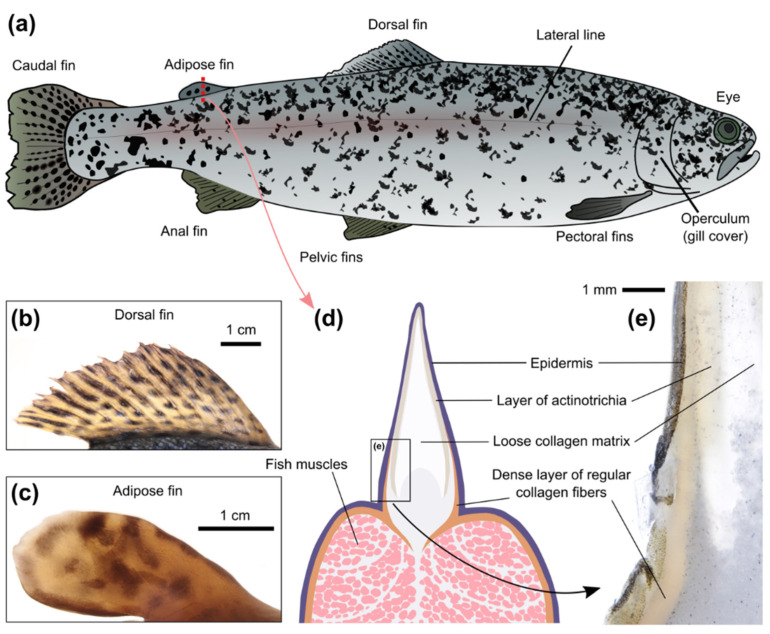
Overview of rainbow trout with focus on the adipose fin. (**a**) General anatomy of the rainbow trout (*Oncorhynchus mykiss*). (**b**) Photo of the dorsal fin of an adult rainbow trout with brightfield illumination. (**c**) Photo of the adipose fin of an adult rainbow trout with brightfield illumination. (**d**) Scheme of the transversal section of the adipose fin and the adjacent muscles. (**e**) A transversal section of the proximal part of the adipose fin. Note the difference in the dermis structures underneath the scaled skin on the back and in the adipose fin.

**Figure 3 animals-12-03042-f003:**
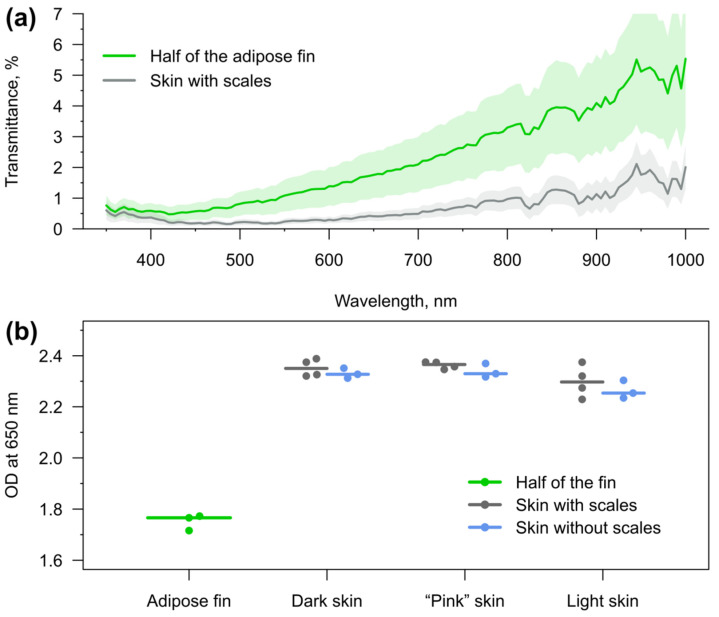
Optical properties of skin and adipose fin of rainbow trout. (**a**) Light transmission through skin and longitudinally cut halves of adipose fin. The spectra are averages over six halves of adipose fin (from three individuals) or 33 skin samples in total from dorsal, lateral, and ventral parts of the fish (from four individuals). The standard deviations are drawn as the ranges. (**b**) Optical density (OD) at 650 nm of the adipose fin and three types of skin with and without scales. Each dot represents individual fish (averaged between two halves of the adipose fin or 2–3 samples for each skin type) and lines show the medians.

**Figure 4 animals-12-03042-f004:**
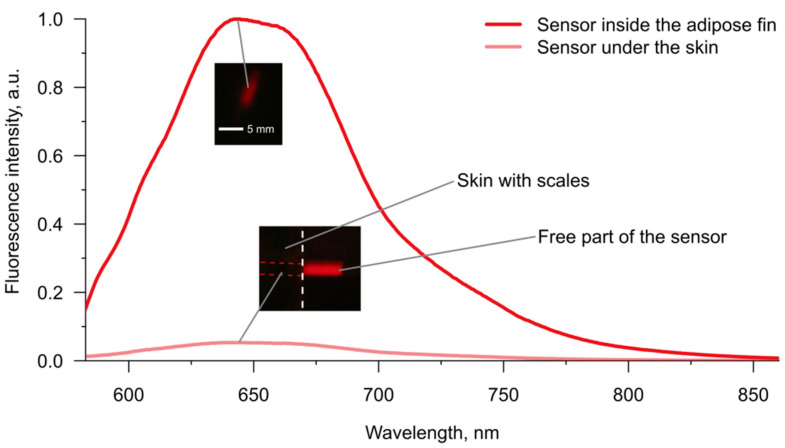
Fluorescence spectra with maximal intensity that were acquired under the microscope from the pH sensor inside the adipose fin or between skin and muscles of the same fish. Fluorescence intensities are normalized to the peak intensity. Large-scale example images of the fluorescent gels were obtained with identical illumination and exposure conditions using a common camera. The gel on the example image was covered with skin, not injected between skin and muscles.

**Figure 5 animals-12-03042-f005:**
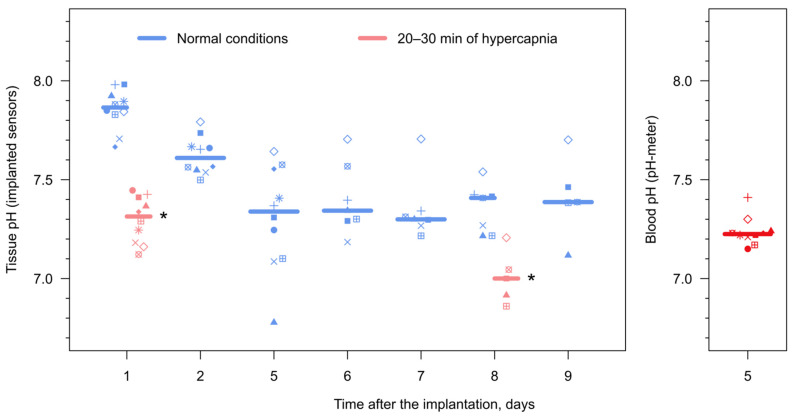
Monitoring tissue pH in the adipose fin of rainbow trout using the implanted sensors in vivo. Each symbol indicates an individual animal, and horizontal lines indicate medians. On days 1 and 8 after implanting the sensors, the animals were subjected to elevated CO_2_ concentration (55–60 mg/L) for 20–30 min right after the measurements in normal conditions. Asterisks indicate statistically significant differences from the pH measurements right before the hypercapnic exposures with *p* < 0.01. Additionally, on day 5, the blood pH was measured right after acquiring the optical signal from the implanted sensors in order to compare the values.

## Data Availability

All the obtained data are available within the article.

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
