# Peer review of "Adipose Fin as a Natural “Optical Window” for Implantation of Fluorescent Sensors into Salmonid Fish"

_animals, 2022, doi:10.3390/ani12213042_

Round 1
Reviewer 1 Report (New Reviewer)
General comments:
This is an interesting topic, and the sensors seem to have a lot of potential benefits for monitoring fish health during times of stress in the hatchery or during transportation. I would have liked to see more information on what kind of sensors are available and what physiological parameters can be monitored by those sensors. Additionally, a description of how the pH sensor specifically works (color change or fluorescence) would be useful as it is unclear how a change in pH is indicated by the sensor. The in vivo experiment appears to have been added since the last round of reviews, and adds to the paper and conclusions overall. However, more information about this experiment is needed in the methods section before results from the experiment are presented in later sections. Please see specific comments below.
Introduction:
This section provides a good introduction to the expanding use of these sensors in humans, mammals, and now fish. However, more information about what physiological parameters can be measured using these sensors and how could be useful. It sounds like the pH sensor was previously tested in zebrafish, and the application seems to be useful for salmonids. However, more information about why the pH sensor was chosen and how it works (maybe more of a method?) would be useful for the reader to understand how and why these sensors might be used in the future for cultured species.
Line 42: not sure what is meant by "cheap implantable sensors to stress markers".
Lines 41-45: This is a long sentence that is a little unclear and hard to follow. Could use rewording.
Line 48: is "the latter" referring only to thin skin, or both fish size and thin skin (both seem applicable in rainbow trout)?
Final paragraph: could expand on your reason for conducting this experiment. Why use sensors in fish? What can be learned from them? What other sensors are or may someday be available? Why did you choose a pH sensor for this experiment? In addition to visualization, is it important that the sensor is also in a location where the fish's physiological response is changing and measurable?
Materials and Methods:
Parts of this section provide some great detail to allow somebody to repeat this experiment. For example, the preparation of the pH sensor and the design of the device used to measure the sensor fluorescence. However, much of the results presented later on in the paper from the in vivo experiment are not well explained in the methods, making it hard to follow what was done.
Section 2.1:
Could use some expansion of the methods. The section clearly describes how tissues were collected, but not what tissues were collected and evaluated. The section mentions "organs", but it seems like this was skin and adipose fin (as opposed to liver or spleen). It is unclear why the tissues were collected. Was this to measure the translucence of the tissues, or were transmitters implanted in the skin and adipose tissues, and the fluorescence of the sensors measured (or both)?
Line 71: What is meant by "transmission of tissues"? Is this light transmission through the tissues (i.e., the color spectrum)? Amount of light coming through the tissues (OD)? Or the transmission of an implanted sensor (not mentioned if this occurred)?
Section 2.2:
Although the preparation of the pH sensor is well explained, it is unclear why a pH sensor was chosen or how a change in pH is measured via this sensor.
Section 2.3:
This section focuses on implantation of the sensor in the adipose tissue, but it seems from the results section that the sensor was put in the skin of live fish as well. What locations in the skin were used? It sounds like the sensor was hard to read through the skin, but since this is a result that is presented later, the methods for implantation and what color of skin was chosen on the live fish is needed here.
Line 120: what is the purpose of the wire? Did the needle not come with a plunger that could be used for injection, so the wire had to be used? Or is there another purpose for the wire?
Lines 128-129: put this sentence in past tense.
Line 131: suggest using a reference to Figure 4 here rather than "see below" (though may require reordering the figures since this would be first mention of a figure and therefore Figure 1).
Line 135: What is "the signal"? A change in color or wavelength? Amount of light? This is where a description of how the sensor works to indicate a change in pH would be useful.
Section 2.4:
All of this appears to be associated with the laser device constructed by the authors (pretty neat device!), but it is unclear whether this device is used to observe the fluorescence in real time with the fish in hand, or if the optical data is recorded and analyzed later. Additionally, the results section mentions the difference between observing fluorescence with the device versus under a microscope, but the microscopic observations are not mentioned in the methods. Did this occur with the tissues from live fish, or was a sensor implanted into the dead fish tissue collected in section 2.1 and examined under a microscope? More detail is needed regarding this topic in either section 2.1 or 2.4.
Results and Discussion:
It is a little hard to tell in what parts of this section results from the experiment performed are presented versus being discussion points from a literature search. Generally in a combined results and discussion section, you present a result from the experiment and then discuss it, but that does not seem to be the case in all subsections. For example, section 3.1 discusses sites for the implantation of optical sensors, but does not appear to present data from the experiment. It appears to be a search of the literature that caused the authors to chose the implant location(s) they did. Some of this may be better suited in an introduction. It may also work to combine subsections 3.1 and 3.2 as 3.2 does appear to present some results from the methods described in section 2.1.
Section 3.1:
Lines 172-176: introduction of a new concept of the color of skin, which is later presented in as results in section 3.2 and Figure 2. This information would be better to include in the methods in section 2.1 when discussing what parts of the skin were collected (and why).
Line 190: mentions a new method that may have been part of the in vivo experiment in which the sensor was implanted in the dorsal fin. This needs to be mentioned in the methods. A full description of the in vivo experiment and what tissues were attempted for implantation prior to settling on the adipose fin would be useful.
Lines 202-204: Are fish less marketable if missing a rayed fin? Even when the fin is purposely removed? Can you provide a citation for this? Additionally, why does the fin need to be removed at all? To prevent human consumption of the sensor?
Lines 227-229: Does this conclusion come from a literature search? Is it based purely on translucence of the fin, or were sensors implanted in other fins? It is hard to tell if any of the other fins were tested in the in vivo experiment. If these fins weren't tested, but the authors think they may be a viable option, maybe add a sentence stating that these fins should be thoroughly tested in future studies.
Section 3.2:
This seems to be the first section to present results from the experiment, those described in section 2.1. Might work to combine sections 3.1 and 3.2 so that the results and discussion work together based on the information collected during the study. This part of the results and discussion makes the assumption that the reader remembers the wavelengths of the light spectrum. It could be useful to remind readers what wavelengths correspond to ultraviolet, blue light, visible light (other colors), and infrared by putting the spectrum under the x-axis in Figure 2a.
Lines 251-253: I don't quite understand the fold math here. Based on Figure 2a the lines appear closer together at the 530-545 range (stated to have a five-fold increase in adipose over skin), but a four fold between 515 and 700 nm. The lines get further apart the higher in the spectrum you go, so it seems like it should be even higher fold at 700 than 515.
Line 255: removal of the scales and the skin color/types tested should be mentioned in the methods section (section 2.1?).
Line 256-259: presents good results based on the skin analysis. However, the major result, that the adipose tissue has a significantly lower OD at 650 nm than any of the skin types is missing - seems like an important result to present not only in the figure, but also in the text.
Section 3.3:
Lines 265-266: Good info about the pH sensor, but it would be useful for this to come earlier in the paper (introduction or methods).
Line 269: first mention of the fluorescent microscope. It is unclear in what part of the experiment described in the methods that this was used. Is everything described in section 3.3 done using the microscope versus the device described in Figure 4? In what part of the experiment, dead tissue or live fish, was the sensor placed between the skin and the meat? Both parts? Put this info in methods as well.
Figure 3: What is the free part of the sensor? This has not been mentioned previously. Is it possible to implant a sensor like this such that a small part sticks out? Seems like you get better fluorescence on the free part than under the skin or under the adipose fin.
Line 283: The last sentence of the figure caption directly contradicts the first sentence that says that the sensor was implanted between skin and meat.
Section 3.4:
Line 300: Didn't know that a sensor had been placed under the skin of live rainbow trout up to this point. Please put this detail in the methods section.
Lines 309-312: It seems like the sensor was effective at measuring the change in pH as was the device at responding to the change in the sensor. However, it is still unclear how the sensor works. Did the senor change color, peak of wavelength, etc.? How did the information you obtained from the sensor correspond to a pH reading? Was there a line or curve to describe the relationship between the information from the sensor and pH such that you could determine pH based on the fluorescence of the sensor?
Section 4:
Lines 322-323: The opening sentence of the conclusion does not appear to be based on data obtained from this experiment, correct? Were other fins tested such that they could be promising alternatives in large fish species, or did this come from the literature? Citation needed?
Author Response
Comments and Suggestions for Authors
General comments:
This is an interesting topic, and the sensors seem to have a lot of potential benefits for monitoring fish health during times of stress in the hatchery or during transportation. I would have liked to see more information on what kind of sensors are available and what physiological parameters can be monitored by those sensors. Additionally, a description of how the pH sensor specifically works (color change or fluorescence) would be useful as it is unclear how a change in pH is indicated by the sensor. The in vivo experiment appears to have been added since the last round of reviews, and adds to the paper and conclusions overall. However, more information about this experiment is needed in the methods section before results from the experiment are presented in later sections. Please see specific comments below.
Authors: Dear Reviewer 1, we would like to deeply thank you for your in-depth review of our manuscript and for the very specific comments provided. Please find our response to each of them below (also as the attached file that may be easier to read).
Introduction:
This section provides a good introduction to the expanding use of these sensors in humans, mammals, and now fish. However, more information about what physiological parameters can be measured using these sensors and how could be useful. It sounds like the pH sensor was previously tested in zebrafish, and the application seems to be useful for salmonids. However, more information about why the pH sensor was chosen and how it works (maybe more of a method?) would be useful for the reader to understand how and why these sensors might be used in the future for cultured species.
Authors: We significantly expanded the Introduction in order to briefly describe the existing implantable optical sensors that can be of interest for fish physiology. The main principles of their functioning are now also explained along with the motivation for choosing the pH sensor (at the very end of Introduction). We would like to thank you for this especially useful recommendation.
Line 42: not sure what is meant by "cheap implantable sensors to stress markers".
Authors: This is now rephrased.
Lines 41-45: This is a long sentence that is a little unclear and hard to follow. Could use rewording.
Authors: It’s done.
Line 48: is "the latter" referring only to thin skin, or both fish size and thin skin (both seem applicable in rainbow trout)?
Authors: Originally, the intention was to refer to thin skin only. But since the size and skin thickness are usually related, we do not see much of a contradiction here.
Final paragraph: could expand on your reason for conducting this experiment. Why use sensors in fish? What can be learned from them? What other sensors are or may someday be available? Why did you choose a pH sensor for this experiment? In addition to visualization, is it important that the sensor is also in a location where the fish's physiological response is changing and measurable?
Authors: We added the information about other implantable sensors a bit earlier and here we explained the choice of pH as the marker. The question of responsiveness of a certain organ to the organism-level reactions is also important of course, and we now touch it deeper a bit later along the Results and Discussion. We would like to thank you for these suggestions.
Materials and Methods:
Parts of this section provide some great detail to allow somebody to repeat this experiment. For example, the preparation of the pH sensor and the design of the device used to measure the sensor fluorescence. However, much of the results presented later on in the paper from the in vivo experiment are not well explained in the methods, making it hard to follow what was done.
Section 2.1:
Could use some expansion of the methods. The section clearly describes how tissues were collected, but not what tissues were collected and evaluated. The section mentions "organs", but it seems like this was skin and adipose fin (as opposed to liver or spleen). It is unclear why the tissues were collected. Was this to measure the translucence of the tissues, or were transmitters implanted in the skin and adipose tissues, and the fluorescence of the sensors measured (or both)?
Authors: We substantially re-wrote the section (currently named «Obtaining tissue samples and light transmission analysis») in order to meet these and other comments.
Line 71: What is meant by "transmission of tissues"? Is this light transmission through the tissues (i.e., the color spectrum)? Amount of light coming through the tissues (OD)? Or the transmission of an implanted sensor (not mentioned if this occurred)?
Authors: Yes, here (current section «Obtaining tissue samples and light transmission analysis») we mostly meant the spectrum of light transmission through tissues. We now changed the term to «transmittance» and added a relevant reference in order to be specific. Specifically for 650 nm we also compared the optical density (OD; in fact, just negative decimal logarithm of the obtained transmittance, i.e. -log10T), so we now clarified it here as well. Furthermore, our comparison of fluorescence intensity for implanted sensors also indirectly demonstrates the difference in tissues’ transmission, so we described this part of the work here as well. We hope the section now became more clear and readable.
Section 2.2:
Although the preparation of the pH sensor is well explained, it is unclear why a pH sensor was chosen or how a change in pH is measured via this sensor.
Authors: Motivation for choosing pH is now added at the end of Introduction, while the working mechanism of SNARF-1 is now explained at the beginning of the section «Preparation and implantation of the pH sensor». Technical details of the pH measurement are presented in the section «Acquiring fluorescence of the sensor and pH measurements».
Section 2.3:
This section focuses on implantation of the sensor in the adipose tissue, but it seems from the results section that the sensor was put in the skin of live fish as well. What locations in the skin were used? It sounds like the sensor was hard to read through the skin, but since this is a result that is presented later, the methods for implantation and what color of skin was chosen on the live fish is needed here.
Authors: Yes, the sensors were injected under skin of alive fish, and this info is now added to the section 2.4. Additionally, we completely re-structured the sections within Methods and added more clear description in order to adhere to these and other concerns.
Line 120: what is the purpose of the wire? Did the needle not come with a plunger that could be used for injection, so the wire had to be used? Or is there another purpose for the wire?
Authors: Yes, the wire was used just as a plunger inside the needle; the used needles did not come with a plunger. It now was made clear in the current section «Preparation and implantation of the pH sensor».
Lines 128-129: put this sentence in past tense.
Authors: It’s corrected now.
Line 131: suggest using a reference to Figure 4 here rather than "see below" (though may require reordering the figures since this would be first mention of a figure and therefore Figure 1).
Authors: We significantly reordered the sections within Methods to make the story more clear, and the explanation to the sensor visualization is now presented before describing the performed analyses. Furthermore, we indeed moved the technical scheme of the hand-hand device to the Methods as the new Figure 1. We would like to thank you for this especially useful recommendation.
Line 135: What is "the signal"? A change in color or wavelength? Amount of light? This is where a description of how the sensor works to indicate a change in pH would be useful.
Authors: In the updated manuscript version we extensively explain the working principle of the sensors within sections 2.1-2.2, so we hope now at this point it should be clear.
Section 2.4:
All of this appears to be associated with the laser device constructed by the authors (pretty neat device!), but it is unclear whether this device is used to observe the fluorescence in real time with the fish in hand, or if the optical data is recorded and analyzed later. Additionally, the results section mentions the difference between observing fluorescence with the device versus under a microscope, but the microscopic observations are not mentioned in the methods. Did this occur with the tissues from live fish, or was a sensor implanted into the dead fish tissue collected in section 2.1 and examined under a microscope? More detail is needed regarding this topic in either section 2.1 or 2.4.
Authors: We tried to clarify all the mentioned details within the current sections «Acquiring fluorescence of the sensor and pH measurements» and «Obtaining tissue samples and transmission analysis». Importantly, we added a starting explanation to the current section 2.2 regarding the specific purposes why each setup was built and used and clearly highlighted the information about the procedure for pH measurements.
Concerning the real-time pH measurements, in our experiments we first acquired the SNARF-1 spectra and then analyzed them a bit later (now clarified in the text) since immediate presenting of the pH data was not required. However, the real-time pH analysis is also totally possible, it’s just a matter of software settings (in our case, writing the necessary script functions in Scilab).
Results and Discussion:
It is a little hard to tell in what parts of this section results from the experiment performed are presented versus being discussion points from a literature search. Generally in a combined results and discussion section, you present a result from the experiment and then discuss it, but that does not seem to be the case in all subsections. For example, section 3.1 discusses sites for the implantation of optical sensors, but does not appear to present data from the experiment. It appears to be a search of the literature that caused the authors to chose the implant location(s) they did. Some of this may be better suited in an introduction. It may also work to combine subsections 3.1 and 3.2 as 3.2 does appear to present some results from the methods described in section 2.1.
Authors: The section 3.1 contains description of our own results of implanting the hydrogels into different parts of rainbow trout and the results of transversal sectioning of the adipose and dorsal fins. However, we agree that these results were a bit difficult to follow among the general information and the information that we cite from the literature here, so we tried to highlight the actual results in the updated version of this section. We suppose that the pieces of general information about fish tissues are essential at this section in order to discuss them jointly with actual results. Moreover, this information is a bit technical and would be excessive in the Introduction. Additionally, we indeed tried to combine sections 3.1 and 3.2 during the revision, but the joint section became quite large and difficult to follow. We thus would like to keep separate sections 3.1 and 3.2 with focuses on their topics and hope the revision made the actual results more clear.
Section 3.1:
Lines 172-176: introduction of a new concept of the color of skin, which is later presented in as results in section 3.2 and Figure 2. This information would be better to include in the methods in section 2.1 when discussing what parts of the skin were collected (and why).
Authors: The information is now moved to the Methods as requested.
Line 190: mentions a new method that may have been part of the in vivo experiment in which the sensor was implanted in the dorsal fin. This needs to be mentioned in the methods. A full description of the in vivo experiment and what tissues were attempted for implantation prior to settling on the adipose fin would be useful.
Authors: We now extensively expanded the section «Obtaining tissue samples and transmission analysis» along with the preceding sections in order to fulfill this and other recommendations. Furthermore, the section 3.1 is now substantially re-written (especially the beginning) and the mentioned details should now be clear.
Lines 202-204: Are fish less marketable if missing a rayed fin? Even when the fin is purposely removed? Can you provide a citation for this? Additionally, why does the fin need to be removed at all? To prevent human consumption of the sensor?
Authors: Yes, the sensor is usually not intended for human consumption (this idea is now added to the text), and removing specifically the sensor and not the whole fin is a bit challenging. We cannot provide a relevant citation, but our impression says that in many cases the fish has to look natural for selling. Since this is of course just an impression, we made the phrase less strong (exchanged “will” with “may”) and removed the related passage from Conclusions.
Lines 227-229: Does this conclusion come from a literature search? Is it based purely on translucence of the fin, or were sensors implanted in other fins? It is hard to tell if any of the other fins were tested in the in vivo experiment. If these fins weren't tested, but the authors think they may be a viable option, maybe add a sentence stating that these fins should be thoroughly tested in future studies.
Authors: In the updated manuscript version we clearly focus the section 3.1 only on the possibilities to inject the sensors into different parts of the fish and on overall structure of the fins. So, this conclusion at the end of the section is made from (i) our results showing easy implantation of the sensors into the proximal part of the dorsal fin (of deceased individuals) and (ii) our transversal sectioning of the adipose fin and the dorsal fin. Additionally, we now mention in the section 3.3 our results of microscoping the semiliquid hydrogel sensor inside the dorsal fin, which is not quantitatively comparable with other data (that’s why not added to the Figure 4), but proves the visibility of the sensor inside the proximal part of the dorsal fin and, thus, its potential as the implantation site.
Section 3.2:
This seems to be the first section to present results from the experiment, those described in section 2.1. Might work to combine sections 3.1 and 3.2 so that the results and discussion work together based on the information collected during the study. This part of the results and discussion makes the assumption that the reader remembers the wavelengths of the light spectrum. It could be useful to remind readers what wavelengths correspond to ultraviolet, blue light, visible light (other colors), and infrared by putting the spectrum under the x-axis in Figure 2a.
Authors: We tried to highlight the actual results in section 3.1 and would like to keep it separate from section 3.2 since the joint section would be too large to follow. Regarding Fig 3a (previous Fig 2a), we believe adding the legend with color spectrum would make the figure too complex. However, we added the necessary landmark wavelengths along the describing text to each mention of any color (including Introduction) in order to make reading the text easier for everyone.
Lines 251-253: I don't quite understand the fold math here. Based on Figure 2a the lines appear closer together at the 530-545 range (stated to have a five-fold increase in adipose over skin), but a four fold between 515 and 700 nm. The lines get further apart the higher in the spectrum you go, so it seems like it should be even higher fold at 700 than 515.
Authors: Here we always compared the average values, i.e. the main spectra presented on Fig. 2a. The wide standard deviations around the average for the adipose fin, of course, indicate relatively high variability and give the wrong impression that the spectra go apart in the higher wavelength, but comparing the average values seems to be the most appropriate choice.
Line 255: removal of the scales and the skin color/types tested should be mentioned in the methods section (section 2.1?).
Authors: It’s now fully explained in the current section 2.3.
Line 256-259: presents good results based on the skin analysis. However, the major result, that the adipose tissue has a significantly lower OD at 650 nm than any of the skin types is missing - seems like an important result to present not only in the figure, but also in the text.
Authors: OD is in fact -log10T, so it’s just another way to present the same information. Since the substantial difference between skin and the adipose fin is obvious from Figure 3a, in the text to Figure 3b we concentrated on the new information, i.e. comparison between skin types and the influence of the scale. The adipose fin is added here mostly as the reference to show that influence of these two factors is negligible.
Section 3.3:
Lines 265-266: Good info about the pH sensor, but it would be useful for this to come earlier in the paper (introduction or methods).
Authors: The specific wavelengths for excitation and fluorescence intensity registration are fully disclosed in the section «Acquiring fluorescence of the sensor and pH measurements» within the Methods.
Line 269: first mention of the fluorescent microscope. It is unclear in what part of the experiment described in the methods that this was used. Is everything described in section 3.3 done using the microscope versus the device described in Figure 4? In what part of the experiment, dead tissue or live fish, was the sensor placed between the skin and the meat? Both parts? Put this info in methods as well.
Authors: All these questions are now thoroughly described within the current section 2.3 in Methods. Briefly, yes, all the data in the section 3.3 are obtained for dead animals under the microscope and only the section 3.4 include results of the experiment with alive fish.
Figure 3: What is the free part of the sensor? This has not been mentioned previously. Is it possible to implant a sensor like this such that a small part sticks out? Seems like you get better fluorescence on the free part than under the skin or under the adipose fin.
Authors: This is now explained at the end of the section 2.3 in Methods. Specifically for imaging purposes (not for the spectrum comparison) we just partially covered the sensor with skin, not implanted between skin and meat. It was made only to visually demonstrate to the readers the dramatic influence of the skin on sensor visibility.
Line 283: The last sentence of the figure caption directly contradicts the first sentence that says that the sensor was implanted between skin and meat.
Authors: This is now clarified at the end of the section 2.3 in Methods.
Section 3.4:
Line 300: Didn't know that a sensor had been placed under the skin of live rainbow trout up to this point. Please put this detail in the methods section.
Authors: It’s now mentioned in the section 2.4.
Lines 309-312: It seems like the sensor was effective at measuring the change in pH as was the device at responding to the change in the sensor. However, it is still unclear how the sensor works. Did the senor change color, peak of wavelength, etc.? How did the information you obtained from the sensor correspond to a pH reading? Was there a line or curve to describe the relationship between the information from the sensor and pH such that you could determine pH based on the fluorescence of the sensor?
Authors: The details of the sensor application are now described in sections «Preparation and implantation of the pH sensor» and «Acquiring fluorescence of the sensor and pH measurements» within Methods. Calibration of I605/I640 intensities to actual pH values is also described in «Acquiring fluorescence of the sensor and pH measurements».
Section 4:
Lines 322-323: The opening sentence of the conclusion does not appear to be based on data obtained from this experiment, correct? Were other fins tested such that they could be promising alternatives in large fish species, or did this come from the literature? Citation needed?
Authors: Yes, the opening sentence in the Conclusions is based on our own results of using the proximal part of the dorsal fin as the injection site. The results are now more clearly mentioned in the sections 3.1 and 3.3.
Submission Date
31 August 2022
Date of this review
15 Sep 2022 05:31:27
Thank you very much again for your time and the suggestions provided!
Very sincerely yours,
Dr. Ekaterina Borvinskaya
Irkutsk State University

Reviewer 2 Report (New Reviewer)
Reviewer comments for Rzhechitskiy et al, ‘Adipose fin as a natural “optical window” for implantation of fluorescent sensors into salmonid fish’, submitted to Animals (Manuscript ID animals-1920157).
The authors describe development of a technique using implantable optical sensors to monitor, in real time, pH in rainbow trout and highlight the utility of the adipose fin as a site of sensor implantation. The manuscript is generally well written and easy to follow, the subject is of great interest to a wide audience, and thus the manuscript seems suitable for publication in the Journal pending response to the five comments below.
First, while I very much recognize that this work represents a ‘first step’ approach toward the development of this protocol in rainbow trout, I’m somewhat surprised that only 4 fish were used in the in vivo part of the study despite the purported “easy implantation and visualization of the sensor”. The authors should describe the factors that limited the study to only 4 fish.
Second, the authors artificially-induced metabolic acidosis to affect and subsequently detect changes in pH, and I would assume this 0.9-unit pH change would be relatively large compared to normal pH fluctuations resulting from natural causes. The authors should cite examples of ‘real-life’ meaningful pH fluctuations and assure the reader that this technology is sensitive enough to detect these differences. Furthermore, the authors should discuss the need to validate their optical sensors. That is, are the pH levels recorded in the adipose fin with these sensors consistent with pH levels recorded from blood or tissue sampling using traditional methods?
Third, in the introduction the authors set forth the potential for implantable sensors to be used on-farm for long-term monitoring of animal health and early identification of infection, but the current study was only conducted for two days. Why weren’t the fish maintained for longer-term pH monitoring? What is the useful life of the pH sensors used in the study? The authors should describe opportunities/limitations to long-term monitoring with this new technology.
Fourth, it isn’t readily apparent to me in section 2.4 what the ‘three different setups’ were in this study. Recommend that the authors revise this section to make it clear to readers with less technical expertise like myself.
And fifth and finally, present results from a means separation test in figure 4C and section 3.4, but statistical analysis methods are not provided in the manuscript. Please revise the manuscript to describe the statistical analysis used.
Author Response
Comments and Suggestions for Authors
Reviewer comments for Rzhechitskiy et al, ‘Adipose fin as a natural “optical window” for implantation of fluorescent sensors into salmonid fish’, submitted to Animals (Manuscript ID animals-1920157).
The authors describe development of a technique using implantable optical sensors to monitor, in real time, pH in rainbow trout and highlight the utility of the adipose fin as a site of sensor implantation. The manuscript is generally well written and easy to follow, the subject is of great interest to a wide audience, and thus the manuscript seems suitable for publication in the Journal pending response to the five comments below.
Authors: Dear Reviewer 2, we would like to thank you for high estimate of our work and the helpful suggestions provided. Please find our response to each point below (also as the attached file that may be easier to read). Please note that we slightly changed the order of your comments to combine the responses to comments #1 and #3.
First, while I very much recognize that this work represents a ‘first step’ approach toward the development of this protocol in rainbow trout, I’m somewhat surprised that only 4 fish were used in the in vivo part of the study despite the purported “easy implantation and visualization of the sensor”. The authors should describe the factors that limited the study to only 4 fish.
Third, in the introduction the authors set forth the potential for implantable sensors to be used on-farm for long-term monitoring of animal health and early identification of infection, but the current study was only conducted for two days. Why weren’t the fish maintained for longer-term pH monitoring? What is the useful life of the pH sensors used in the study? The authors should describe opportunities/limitations to long-term monitoring with this new technology.
Authors: Unfortunately some co-authors got COVID-19 just before the first submission deadline and we had to include only a small experiment into the old version of the manuscript. Since we now had more time, we performed another experiment from scratch for the updated manuscript version. It included 10 individuals that were monitored for 9 days and also showed good applicability of the sensors. To our surprise, the new experiment showed lower median pH one day after the injection: 7.9 instead of 8.5. However, in the long-term monitoring the tissue pH decreased and leveled off at about 7.3-7.4, which was found to be in agreement with the blood pH measurements (using a pH-meter). It seems like the process of pH decrease was going on in both experiments, but the velocity could be different and this explains the observed difference in starting pH. The potential causes of the decrease are now discussed in the manuscript.
Regarding the useful lifetime of the implantable sensors, we can imagine two issues: (i) immune foreign response to the implant and (ii) photobleaching of the fluorescent component. We currently perform another study with focus on the first issue and planning to prepare a separate full paper concerning the long-term biocompatibility. However, from the second experiment we now can say that the sensors are still in contact with tissues for at least 8 days after injection. We also didn’t observe significant photobleaching of the sensors during the experiment, but cannot present any additional quantitative data. The limitations are now mentioned in the section 3.4.
Second, the authors artificially-induced metabolic acidosis to affect and subsequently detect changes in pH, and I would assume this 0.9-unit pH change would be relatively large compared to normal pH fluctuations resulting from natural causes. The authors should cite examples of ‘real-life’ meaningful pH fluctuations and assure the reader that this technology is sensitive enough to detect these differences. Furthermore, the authors should discuss the need to validate their optical sensors. That is, are the pH levels recorded in the adipose fin with these sensors consistent with pH levels recorded from blood or tissue sampling using traditional methods?
Authors: We now added the comparison of pH measurement with the implanted sensors inside the adipose fin and using a pH-meter with a microelectode for extracted blood, and they seem to be consistent enough. We also now cite additional examples of previously observed blood pH changes under different conditions as suggested.
Fourth, it isn’t readily apparent to me in section 2.4 what the ‘three different setups’ were in this study. Recommend that the authors revise this section to make it clear to readers with less technical expertise like myself.
Authors: We added a starting explanation to the section (its number is 2.2 now) regarding the specific purposes why each setup was built and used. We hope the section now became more clear.
And fifth and finally, present results from a means separation test in figure 4C and section 3.4, but statistical analysis methods are not provided in the manuscript. Please revise the manuscript to describe the statistical analysis used.
Authors: We now clarified the performed statistical analysis at the very end of the current section 2.4.
Submission Date
31 August 2022
Date of this review
23 Sep 2022 21:55:44
Thank you very much again for your time and the suggestions provided!
Very sincerely yours,
Dr. Ekaterina Borvinskaya
Irkutsk State University

Round 2
Reviewer 1 Report (New Reviewer)
The authors did a good job of addressing reviewer comments and questions. Well done!
This manuscript is a resubmission of an earlier submission. The following is a list of the peer review reports and author responses from that submission.
Round 1
Reviewer 1 Report
This paper reports an optimal choice, the adipose fin, for implantation sites for fluorescent sensors, which is interesting but not very suitable for the journal.
(1) The introduction did not provide sufficient background to let readers know of the specific roles of optical/fluorescent sensors in fish. In addition to marking individuals, what physiological parameters can be monitored? Why does it make sense to monitor these parameters? etc.
(2) The research design is far away from appropriation, and not enough to prove the conclusion. For example, there is no experimental design for how different tissues affects the performance of the fluorescence detection PH or other physiological parameters.
(3) The results are one-sided and inadequate.
Reviewer 2 Report
Summary: In this study, Rzhechitskiy et al. validate the adipose fin as an implant site for optical sensors in salmonid fish. The motivation for this study is to enable monitoring of fish health in aquaculture applications. The authors characterize the transmission spectra of the adipose fin and compare it to the transmission spectra of fish skin. The authors report that at low wavelengths (UV to blue), the transmission spectra of the adipose fin and fish skin are similar, but at longer wavelengths, the transmission coefficients for the adipose fin become significantly higher than that of fish skin. The authors suggest that the differences in tissue composition between the adipose fin and fish back explain the transmission spectra results. The authors then test the potential for optical pH sensing by injecting a fluorescent pH sensor into the amputated adipose fin and measuring the pH.
Comments:
This is an interesting manuscript and has value, but the scope of the study is limited due to the lack of in-vivo data. While the claim that the adipose fin is a good target for communicating with fluorescent sensors is validated by the transmission data, it is unclear if the adipose fin is a good target for the sensors to begin with. The authors state that the fin has a functional purpose which “implies good blood supply”. Does that imply, however, that the fin can be a good target for determining the overall physiological state of the fish? Do the authors have any data that show they can detect disease using a pH sensor in the adipose fin? The pH measurement is a good step in this direction, but more rigorous tests will greatly strengthen this paper. What diseases do the authors think are important to be able to test for? I understand that the main point of this paper is to demonstrate that the adipose fin is a good location for optical sensors, but a more thorough discussion of applications is necessary.
I am also curious about the long-term tissue response to an implanted sensor in the adipose fin. In mammals, it is common for scar tissue to grow around implants, and the contents of this scar tissue is typically collagen. The authors claim that the transparency of the adipose fin is due to the lack of dense collagen fibers – after insult to the fin due to surgery, will the optical properties of the fin change? What is the duration the sensors need to be active for in the fish body?
Finally, the stated goal of this work to automate testing of fish health using wireless sensors. The authors explicitly state in the introduction of their paper “… numbers of these animals are too high to control the status of each individual regularly”. However, isn’t this work reliant on surgically implanting a fluorescent sensor into each individual fish? How is sensor implantation multiplexed? How will data acquisition be multiplexed? Optical readings are often very sensitive to motion which suggests that the fish will need to be immobilized for accurate measurements. While implementation of these methods would be out of the scope of the paper, I do believe they should be discussed. As it stands, I do not understand the logic of using implantable fluorescent sensors to obtain population information.
Additionally, I have some specific concerns with the content and language in the paper that should be addressed before this paper can be deemed appropriate for publication.
Pg 1 ln 27: The grammar is a bit strange. I suggest: “…novel implantable sensors have the potential to revolutionize modern medicine and veterinary practices…”
Pg 1 ln 26-29: This sentence is also very long. It would improve readability if the sentence were split into two. “…sensors have the potential to revolution modern medicine and veterinary practice by providing constant real-time monitoring of physiological parameters. Previous, these could only be measured with blood or tissue samples”
Pg 1 ln 29: I think I understand what the authors are trying to say: traditional EM communication requires a radio, etc. so microelectronics are necessary whereas light based communication can be simpler, but the way this is written is a bit confusing. Furthermore, there exist many sophisticated light-based sensors that do rely on integrated circuits to interpret sensed light data or to modulate light frequency/phase/amplitude. It ultimately depends on the communication protocol employed. For accuracy, I would suggest rewriting this statement to “Implantable sensors using light as a readout can be constructed easily using standard biochemical techniques, but…”
Pg 2 ln 38-41: The sentiment of this sentence is understandable, but the grammar is incorrect which makes the specific information difficult to understand. I am not sure what exactly is meant by “procedures as transportation associated with decrease in water quality”. Is this just trying to say that the water quality the fish are kept in is reduced during transportation or is this suggesting that transportation increases the probability of infectious diseases? Please clarify.
Pg 4 Section 3.1: I think this section of the paper is quite interesting and valuable, but there is no data to support any of these claims. I understand that most of the claims made here are qualitative, but if quantitative data can be provided, that would greatly strengthen the claims here. For example on Pg 4 Ln 141-144: inclusion of data that shows the time constant for equilibration would be very useful. Additionally some diagram or picture of the different locations on the fin would be very helpful to visualize the potential implantation targets.
Pg. 4 Ln 175: For readability, I suggest the transmission spectra data (Fig. 1c) be separated from figure 1 and put closer to this section as a new figure.
Pg 4 Ln 175: What are the thicknesses of the tissue samples used in this study? Is the thickness of skin used here similar to the thickness of the fin?
Pg. 5 Ln 205,206: I’m not sure what is meant by “this dye has both excitation and emission peaks in the range of three-fold higher fin transmission”. I assume this is talking about the relative transmittivity differences between the fin vs the skin? If so, I suggest the wording should be more explicit: “this dye has excitation and emission peaks in the [wavelength range], which as we showed in Figure 1c, has 3-fold higher transmission in the adipose fin than skin”.
Pg 6 Ln 222: Do you have any ground truth measurement to validate the optical measurements taken here?
Pg 6 Ln 224: The validation the authors used in this study was to measure the pH of an amputated fin – the authors suggest that the value they measured (~6.6) is realistic, but they do not have any citation to back up this claim.
Figure 1c: Why is the median of the data plotted instead of the average? Standard deviation should also be plotted along with the mean.
Figure 2: Caption says “search for the reason” - I assume this should say “region”
Figure 2a: Do the authors have any explanation for why the data collected from skin samples has low variance but the data from the adipose fin has one very large outlier? Perhaps more data can be measured to get a better sense of the spread. This is a minor concern as the sensors used in this study are ratiometric, but I am wondering if there is some known reason for this discrepancy (e.g. measurement error, biological variation, etc.)
Figure 2b: It is difficult to understand what is being shown here. To make things clearer, I suggest an accompanying image of the whole fish from the top down with a line showing where this cross section is taken through.